# CRoPS: A Training-Free Hallucination Mitigation Framework for Vision-Language Models

**Neeraj Anand**[1]*, **Samyak Jha**[1], **Udbhav Bamba**[2], **Rahul Rahaman**[3]
[1] *Indian Institute of Technology (ISM), Dhanbad, India*
[2] *Transmute AI*
[3] *National University of Singapore*

**Reviewed on OpenReview:** `https://openreview.net/forum?id=KQSoZDPVGX`

## Abstract

Despite the rapid success of Large Vision-Language Models (LVLMs), a persistent challenge is their tendency to generate hallucinated content, undermining reliability in real-world use. Existing training-free methods address hallucinations but face two limitations: (i) they rely on narrow assumptions about hallucination sources, and (ii) their effectiveness declines toward the end of generation, where hallucinations are most likely to occur. A common strategy is to build hallucinated models by completely or partially removing visual tokens and contrasting them with the original model. Yet, this alone proves insufficient, since visual information still propagates into generated text. Building on this insight, we propose a novel *hallucinated model* that captures hallucination effects by selectively removing key text tokens. We further introduce *Generalized Contrastive Decoding*, which integrates multiple hallucinated models to represent diverse hallucination sources. Together, these ideas form **CRoPS**, a training-free hallucination mitigation framework that improves CHAIR scores by 20% and achieves consistent gains across six benchmarks and three LVLM families, outperforming state-of-the-art training-free methods.[1]

## 1 Introduction

Recent advances in large vision-language models (LVLMs) have demonstrated remarkable multi-modal capabilities (Liu et al., 2023c; Dai et al., 2023). Their ability to comprehend both images and language has enabled a wide range of applications. However, similar to Large Language Models (LLMs), LVLMs are prone to "hallucinations" generating convincing responses that lack precision, which can lead to misleading information (Rani et al., 2024; Li et al., 2016). This limitation poses a significant challenge to their reliability as trustworthy AI assistants in real-world applications (Wang et al., 2023b; Liu et al., 2023a).

Recent works aiming to mitigate hallucinations can be categorized into two regimes: *training-based methods* (Biten et al., 2022; Kim et al., 2023; Rohrbach et al., 2018), and *training-free methods* (Wei et al., 2022; Li et al., 2023; Leng et al., 2024; Wang et al., 2024b; Favero et al., 2024). In this study, we are interested in a very specific group of training-free approaches that utilize a common framework of Contrastive Decoding (CD) (Li et al., 2023). CD, a training-free approach, proposes to *negate* the hallucinations from LLM outputs by contrasting (subtracting) outputs of heavily hallucinated models, essentially reducing the problem to identifying and using hallucinated models to contrast with. These methods often suffer from several shortcomings. Firstly, by design, the hallucinated models used for contrasting cease to perform well as the generation of output tokens (Figure 3), making them effective primarily in the early stages of generation.Furthermore, a single hallucinated model does not sufficiently represent all the sources of hallucination, e.g., hallucination due to uninformative visual tokens and hallucinations from the bias produced by the training data.

---

*Corresponding Author: neerajanandfirst@gmail.com
[1] Code is available at https://github.com/ubamba98/CRoPS-Mitigate-Hallucinations-in-Vision-Language-Models

In this work, we start with an analysis of the existing methods (Favero et al., 2024; Huo et al., 2024) and show, by means of both empirical analysis and theoretical computations, how existing methods struggle to remove hallucination as the LVLM generates more tokens. We then propose a novel hallucinated model that overcomes this challenge by going beyond the removal of full or partial visual tokens and considering textual inputs for removal. Our proposed model is motivated by what our analysis shows, that in the later stages of the generation, previously generated output tokens carry equal, if not more, importance than visual tokens.

Next, we empirically show how existing contrastive decoding methods fail to address different kinds of sources of hallucination. To mitigate this issue, we generalize the concept of contrastive decoding and extend it to accommodate multiple models to contrast with rather than a single model. Our proposed *Generalized Contrastive Decoding*, allows us to conjoin our newly proposed hallucinated model with existing works Huo et al. (2024), to devise our novel, training-free hallucination mitigation method CRoPS (fig. 1). We evaluate and compare our proposed CRoPS against competing methods in a wide range of generative and discriminative benchmarks. We also show that CRoPS outperforms all methods across three different choice of architectures.

**Contribution:** Below we summarize our main contributions. In this work,

- We provide detailed analysis of how existing methods perform poorly in the later stages of the LVLM token generation. This also enables us to devise a novel hallucinated model by moving beyond visual tokens to identify and remove textual inputs from prompt and past generated tokens.

- We empirically show how the individual methods only tackle a specific source of hallucination and fail to remove other sources. To this end, we formulate Generalized Contrastive Decoding, by extending the scope of contrastive decoding to allow multiple models to be contrasted with.

- Finally we combine our novel text-deficit hallucinated model with image-deficit hallucinated model under the novel generalized contrastive decoding framework and propose CRoPS. Our proposed method significantly outperforms baseline, and brings consistent improvement over competing methods across a wide range of tasks, datasets, and LVLM architectures.

## 2 Related Works

With the progress in LLMs, recent studies have explored LVLMs by integrating visual encoders into pre-trained LLMs. These models have shown some advanced multi-modal capabilities. However, they suffer from hallucinations (Rohrbach et al., 2018; Zhang et al., 2024; Guan et al., 2024; Wu et al., 2024b), which restrict their real-world applications (Wang et al., 2023b; Liu et al., 2023a). There are several causes of hallucinations, including a lack of understanding of world knowledge, overfitting to specific training data patterns, and insufficient common sense reasoning. In LLMs, hallucinations typically occur when generated responses contradict real-world knowledge or common sense. In contrast, for VLMs, the main concern is whether the generated response conflicts with the provided image.

Researchers have explored several methods to mitigate this issue, which can be broadly categorized into the following two groups:

**Training-based Approaches.** Training-based methods involve either retraining VLMs with curated datasets or using auxiliary models to supervise or revise generations. These approaches include instruction fine-tuning on hallucination-aware datasets (Lee et al., 2022; Gunjal et al., 2024; Zhao et al., 2024; Jiang et al., 2024; Yu et al., 2024; Yue et al., 2024) and post-hoc training of revisor networks that analyze the model outputs and correct hallucinations using auxiliary networks (Manakul et al., 2023; Zhou et al., 2024; Yin et al., 2024; Chen et al., 2024; Wu et al., 2024a; Feng et al., 2024). While these techniques can be effective, they require extensive computational resources and careful dataset design.

**Training-free Approaches.** Training-free approaches, in contrast, modify inference-time decoding or attention patterns without additional training or supervision. They include decoding-time modifications

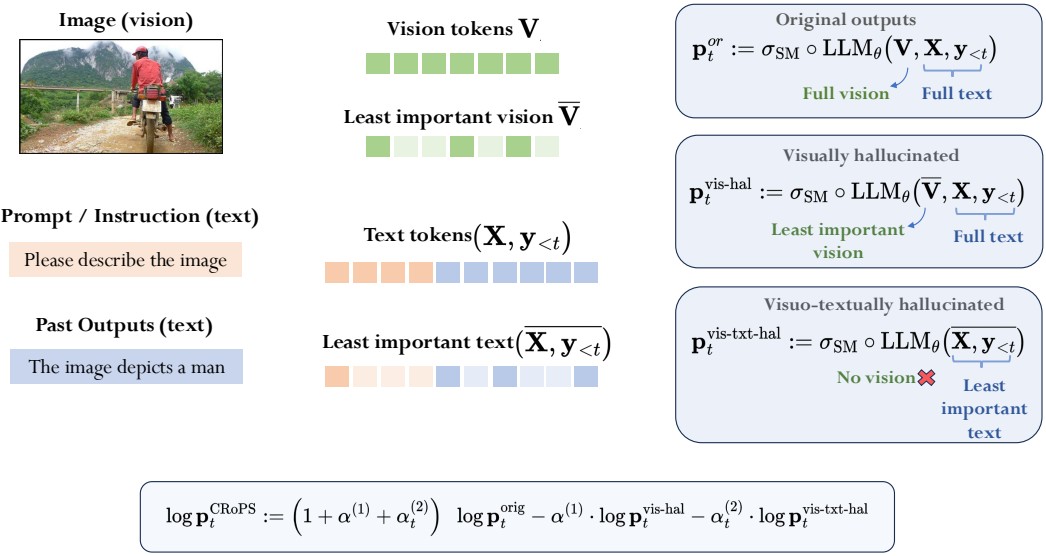

Figure 1: **Overview of CRoPS framework.** CRoPS combines two hallucinated models: one that removes visual tokens to capture vision-related hallucinations, and another that removes key textual tokens to capture text-related hallucinations. Their outputs are then integrated through generalized contrastive decoding framework to reduce hallucinations in LVLMs.

(Huo et al., 2024; Wang et al., 2023b; Favero et al., 2024; Li et al., 2023; Chuang et al., 2024; Liu et al., 2024b; Leng et al., 2024; Wang et al., 2024b; Kim et al., 2024b; Zhu et al., 2024; Huang et al., 2024; Kim et al., 2024a; Woo et al., 2024; Zheng et al., 2024; Li et al., 2025) and attention adjustment mechanisms (Tang et al., 2025; Yin et al., 2025). These methods are simple plug-and-play modules and avoid the computational overhead of training-based methods.

Closely related to our work are Multi-Modal Mutual-Information Decoding (M3ID) (Favero et al., 2024) and Self Introspective Decoding (SID) (Huo et al., 2024), which also fall under the category of training-free approaches. M3ID proposes to utilize and tweak the same LLM to come up with a hallucinated model. They argue that if the image input is removed from the LLM, the model generates arbitrary text that is entirely unrelated to the visual content, relying solely on biases learned during training. SID, on the other hand, selectively retains the least important vision tokens, identified as those exhibiting low coherency with the query token, instead of removing all visual tokens from the LVLM input. The authors argue that these low-importance tokens contribute most to hallucinations. These methods form the foundation of our contrastive hallucination framework, which we explore further in Section 4 (see Appendix A and B for formal definitions).

Unlike prior methods, which independently address visual token dilution and biases introduced during pre-training, *CRoPS* effectively mitigates these issues without requiring fine-tuning or auxiliary models.

## 3 Background

In this section, we formalize LVLM inference and decoding because these details determine how visual information propagates into generated tokens (Section 3.1). We then describe Contrastive Decoding (CD) framework for reducing hallucinations in LVLM (Section 3.2) and an attention-based token pruning method that we use to construct hallucinated models (Section 3.3).

### 3.1 Inference and Decoding in LVLMs.

LVLMs accept both visual and textual input to generate textual output. The model first encodes the input image into a sequence of vision tokens using a vision encoder and a cross-modal projection module. We denote these visual tokens as $\boldsymbol{V} := (\boldsymbol{v}_1, \boldsymbol{v}_2, \ldots, \boldsymbol{v}_m)$, where $m$ is the total number of visual tokens. Additionally, the model receives a textual prompt $\boldsymbol{X}$, which can also be represented as a sequence of $n$ tokens $\boldsymbol{X} = (\boldsymbol{x}_1, \boldsymbol{x}_2, \ldots, \boldsymbol{x}_n)$. The LLM, parameterized by $\theta$, then generates textual output tokens sequentially. Let $\boldsymbol{y}_{<t} := (\boldsymbol{y}_1, \boldsymbol{y}_2, \ldots, \boldsymbol{y}_{t-1})$ be the sequence of generated tokens till inference time step $t$. The inference process can be formally written in the following way:

$$\boldsymbol{p}_t = \text{softmax} \circ \text{LLM}_\theta(\boldsymbol{V}, \boldsymbol{X}, \boldsymbol{y}_{<t}) \tag{1}$$
$$\boldsymbol{y}_t = \text{Decode}[\boldsymbol{p}_t]$$
$$\boldsymbol{Y}_{<t+1} = [\boldsymbol{y}_{<t} : \boldsymbol{y}_t]$$

Initially, the language model outputs logits, denoted by $\text{LLM}_\theta(\cdot)$, which are then converted into probability vector $\boldsymbol{p}_t$ by applying the softmax function. The produced $\boldsymbol{p}_t$ is a probability vector of shape $|\mathcal{V}|$, where $\mathcal{V}$ represents the vocabulary of the LLM. Next, a decoding function is applied to convert $\boldsymbol{p}_t$ into a single token $\boldsymbol{y}_t$. To perform this transformation one of the several available decoding strategies such as greedy, beam search and nucleus sampling can be utilized. The newly generated output token $\boldsymbol{y}_t$ is then appended to the past tokens to provide as input to the LLM for generating the next token.

### 3.2 Hallucination and Contrastive Decoding

An LVLM is said to hallucinate when the generated sequence $\boldsymbol{Y}$ is fully or partially inconsistent with the visual input $\boldsymbol{V}$. As discussed in section 2, several training-free approaches have been proposed to mitigate hallucinations. In this work, we focus on a particular set of methods that share a common framework called *Contrastive Decoding (CD)* (Li et al., 2023). The CD framework tries to remove hallucinations from a LVLM, by suppressing the probabilities assigned to hallucinated tokens / objects. It does so by first creating or identifying a hallucinated model and then subtracting the logits of the hallucinated model from the original LVLM logits, thus reducing the probabilities assigned to the hallucinated tokens. Formally, under the CD framework, the final probability outputs are computed by,

$$\log \boldsymbol{p}_t = (1 + \alpha) \cdot \log \boldsymbol{p}_t^{orig} - \alpha \cdot \log \boldsymbol{p}_t^{hal}, \tag{2}$$

where $\alpha > 0$, $\boldsymbol{p}_t^{orig}$ is the probability outputs from the original LVLM with complete input as computed in equation 1, and $\boldsymbol{p}_t^{hal}$ are the probability outputs from the hallucinated model. As CD relies on defining a hallucinated model, the choice of such a model remains crucial.

### 3.3 Attention-based Token Pruning

At inference time $t$, our input to the LVLM is the tuple $(\boldsymbol{V}, \boldsymbol{X}, \boldsymbol{y}_{<t})$ of visual tokens $\boldsymbol{V}$, text tokens $\boldsymbol{X}$, and the LLM generated outputs so far $\boldsymbol{y}_{<t}$. To estimate token importance, we utilize the attention weights produced by a selected transformer layer $l$ within the language model. Let $\mathbf{K} \in \{\boldsymbol{V}, \boldsymbol{X}, \boldsymbol{y}_{<t}\}$ denote the set of keys and $\boldsymbol{y}_t$ the current query token. Let $\mathbf{K} \in \{\boldsymbol{V}, \boldsymbol{X}, \boldsymbol{y}_{<t}\}$ denote the set of key tokens and $\boldsymbol{y}_t$ the current query token. For a multi-head attention mechanism with $H$ heads, we define the importance score $\psi$ for each key token as the mean attention weight across all heads

$$\psi(\boldsymbol{y}_t) = \frac{1}{H} \sum_{h=1}^{H} \text{Attention}^{(l,h)}(\mathbf{K}, \boldsymbol{y}_t), \tag{3}$$

where $\text{Attention}^{(l,h)}(\mathbf{K}, \boldsymbol{y}_t)$ denotes the attention distribution from the $h^{th}$ head of layer $l$, measuring how strongly the query token $\boldsymbol{y}_t$ attends to the keys in $\mathbf{K}$.

Based on $\psi(\boldsymbol{y}_t)$, we identify the tokens with the lowest importance values, those least attended by the current query. Using this score, we can choose the bottom $\bar{u}(< u = |\mathbf{K}|)$ tokens with lowest $\psi$ scores and

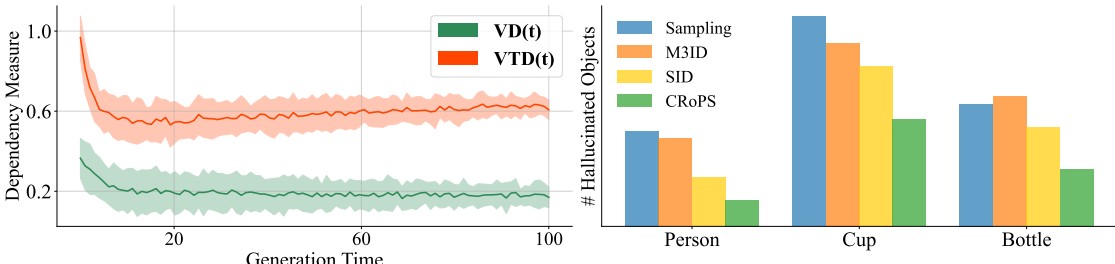

Figure 2: **Left: Plot of dependency measure** (see section 4.1), which quantifies the influence of vision and vision+text tokens on LVLM generation.We observe that VD($t$) decreases over time, indicating that the model relies less on vision tokens as decoding progresses. **Right: Frequency of hallucinated objects** that frequently co-occur with the ground truth object "dining table". We observe that SID and *CRoPS* effectively mitigate statistical biases, whereas M3ID performs sub-optimally.

remove high-importance tokens, to create a new sparser set of tokens. This design is motivated by prior findings from SID (Huo et al., 2024), which demonstrate that low-importance tokens are more likely to induce hallucinations.

## 4  Motivation

In this section, we highlight the limitations of existing contrastive decoding based approaches for hallucination mitigation, which motivate the design of our proposed framework.

### 4.1  Drawback I: Diminishing Dependency on Visual Tokens

To support the following discussion, we first define a metric that captures how LLM generation behavior depends on the input as a function of generation time $t$. Specifically, we extend the *Prompt Dependency Measure* proposed in Favero et al. 2024 to define two new metrics: *Visual Dependency* and *Visuotextual Dependency* as given in equation 4 and equation 8 respectively. The *Visual Dependency* is defined as,

$$\mathrm{VD}(t) := \mathrm{dist}\big(\mathrm{softmax} \circ \mathrm{LLM}_\theta\big(\boldsymbol{V}, \boldsymbol{X}, \boldsymbol{y}_{<t}\big), \mathrm{softmax} \circ \mathrm{LLM}_\theta\big(\boldsymbol{X}, \boldsymbol{y}_{<t}\big)\big) \tag{4}$$

where $\mathrm{dist}(P, Q) = \frac{1}{\sqrt{2}}\left\|\sqrt{P} - \sqrt{Q}\right\|_2$ represents the hellinger distance. VD(t) measures the change in generated output token distribution if the visual input tokens are omitted. The left subfigure of Figure 2 plots VD($t$) as a function of generation step $t$. It is quite evident that VD($t$) decreases drastically as we generate more tokens, i.e., the distribution shift when we ignore the image is less during the later stage of the generation.

This diminishing dependency has important implications for contrastive decoding methods, as it directly affects the effectiveness of contrastive signals used during later stages of generation. Since Favero et al. 2024 defines their hallucinated model as $\boldsymbol{p}_t^{hal} := \mathrm{LLM}_\theta\big(\boldsymbol{X}, \boldsymbol{y}_{<t}\big)$, a diminishing VD($t$) points to increasing similarity between $\boldsymbol{p}_t^{orig}$, and $\boldsymbol{p}_t^{hal}$ over time (derived in Appendix E). As a result, the contrastive signal becomes less meaningful in the later stages of generation, rendering the hallucinated model less effective (as also qualitatively illustrated in Figure 3, where later-stage generations exhibit stronger hallucinations). Similarly, the masking of image tokens in the method by Huo et al. 2024 becomes less impactful when VD($t$) itself is low.

### 4.2  Remedy I: A Novel Model to Contrast with

As the diminishing VD($t$) indicates increasing redundancy of visual tokens, later-stage generation primarily depends on textual context. Thus, hallucinated models that partially or fully discard visual tokens lose effectiveness in the later stages. To alleviate this issue, we propose removing important textual tokens,

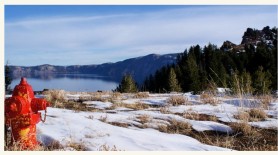

Prompt: Please describe this image in detail

Vanilla:- The image showcases a beautiful snowy landscape with mountains, grass-covered hills, and a valley situated between them. It's a serene scene featuring a lake nestled within the valley. Snow-covered mountains rise dramatically into the sky, with the water below providing a picturesque contrast to the snow-covered terrain. There are several trees in the image, some of which are bare and have snow on them, while others are covered in snow and appear to be surrounded by the lake. The snowy landscape creates an idyllic, natural setting perfect for hiking, skiing, or simply enjoying the majestic beauty of the winter season.

SID:- The scene features a snowy hill overlooking a body of water, possibly a lake. In the background, there is a mountain or mountain range providing a picturesque view. The snow covers both the hill and the ground, creating a serene atmosphere around the lake. There are several trees in the background, with some in the foreground as well. Some of these trees are partially hidden in the snow. In the sky, a cloud can be seen, adding to the wintry atmosphere. Additionally, a horse is present near the edge of the scene, enhancing the picturesque nature of the location.

M3ID:- The image showcases a serene winter landscape, with a picturesque hillside overlooking a large body of water. The snow-covered hillside is surrounded by a forest filled with pine trees, creating a sense of tranquility. The mountains and grassy ground with tree branches add to the beautiful natural scenery. A cow can be seen grazing near the rocky field in the foreground, sharing the landscape with the cow as nature remains alive during the snowy winter season.

CRoPS:- The image features a scenic landscape with mountains surrounding a large body of water, likely a lake or a river. The mountains stretch across the background, creating a picturesque scene. Trees can be seen scattered throughout the area, adding to the natural beauty of the landscape. The sky above is mostly clear with soft streaks of clouds. The snow on the ground creates a serene atmosphere, perfect for winter activities such as skiing or snowboarding. The presence of mountains and water creates a sense of tranquility and adventure in this winter wonderland.

Figure 3: Comparison of image descriptions from different methods. Vanilla, SID, and M3ID contain hallucinated details (highlighted in red), e.g., animals and exaggerated snow coverage. In contrast, CRoPS produces a faithful description without these hallucinations. Note that hallucinations become more frequent during later stages of generation.

where the importance score is computed from the attention weights at an early layer $l$ of the model, similar to the strategy adopted by SID (Huo et al., 2024) for visual tokens.

$$\psi(\boldsymbol{y}_t) = \frac{1}{H} \sum_{h=1}^{H} \text{Attention}^{(l,h)}\big([\boldsymbol{X}, \boldsymbol{y}_{<t}], \boldsymbol{y}_t\big) \tag{5}$$

Using these importance scores, we only retain the least important text (prompt and past tokens combined) tokens and convert the tuple $(\boldsymbol{X}, \boldsymbol{y}_{<t})$ into,

$$\overline{\boldsymbol{X}, \boldsymbol{y}_{<t}} := \text{LeastImp}\Big[(\boldsymbol{X}, \boldsymbol{y}_{<t}), \eta(\mu, t)\Big]. \tag{6}$$

by keeping $\eta(\mu, t)$ text tokens where $\eta(\mu, t) = \beta_0 + \beta_1\big(1 - e^{-\mu t}\big)$ is a non-decreasing function with respect to $t$. Since the number of text tokens increases with time (past generated tokens), $\eta(\mu, t)$ needs to be a non-decreasing function to maintain sparsity of the retained incoherent tokens. Discussed in detail in section 7.

Our proposed hallucinated model then takes the form,

$$\boldsymbol{p}_t^{\text{vis-txt-hal}} := \text{softmax} \circ \text{LLM}_\theta\big(\overline{\boldsymbol{X}, \boldsymbol{y}_{<t}}\big), \tag{7}$$

This means we not only completely remove visual tokens, but we also remove important textual tokens. To check if the proposed hallucinated model is free from the problem of diminishing dependency, we compute *Visuotextual Dependency* VTD($t$) as,

$$\text{VTD}(t) := \text{dist}\big(\text{softmax} \circ \text{LLM}_\theta\big(\boldsymbol{V}, \boldsymbol{X}, \boldsymbol{y}_{<t}\big), \text{softmax} \circ \text{LLM}_\theta\big(\overline{\boldsymbol{X}, \boldsymbol{y}_{<t}}\big)\big) \tag{8}$$

and compare against VD($t$) in the left subfigure of Figure 2. It is evident that unlike VD($t$), VTD($t$) does not diminish as time passes. Meaning, the proposed hallucinated model significantly differs from the original outputs and hence contrasting is effective even in the later stages of generation.

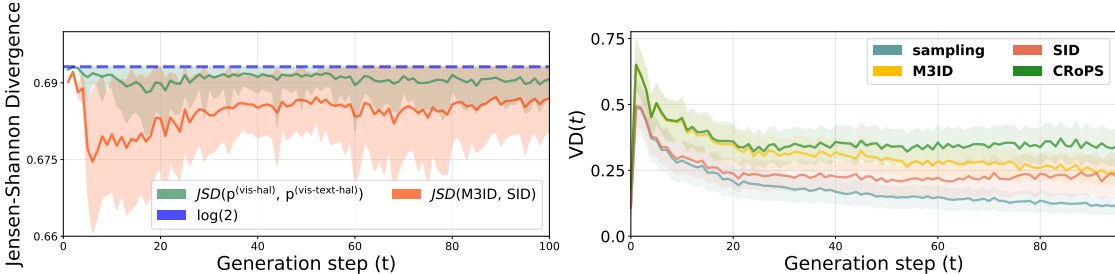

Figure 4: **Left: Plot of Jensen-Shannon (JS) divergence** over generation time between different hallucinated models. The dashed blue line indicates log(2), which is the maximum possible divergence value. **Right: Plot of Visual Dependency** of final outputs across different methods (Sampling, SID, M3ID, and CRoPS).

### 4.3 Drawback II: Contrasting with A Single Hallucinated Model is Insufficient

Contrastive decoding methods typically rely on hallucinated models that either ignore or perturb parts of the input to expose hallucinations. These models are designed to capture different failure modes such as over-reliance on past generated tokens or attention to irrelevant visual features. Although such methods show promising results, addressing only one of these sources of hallucination is insufficient.

For instance, M3ID removes the visual input entirely, thereby targeting hallucinations arising from excessive dependency on text tokens or language priors. However, it fails to mitigate hallucinations triggered by misleading or spurious visual cues. In contrast, SID builds a hallucinated model using low-importance visual tokens and contrasts it with the original model to suppress hallucinations caused by irrelevant visual cues. However, it remains ineffective against hallucinations that emerge later in generation when textual dependency dominates.

As discussed in section 4.1 and depicted by the decreasing Visual Dependency VD($t$) in Figure 2 (left), the influence of visual tokens diminishes over time, reducing the impact of methods that modify only visual inputs. Furthermore, Figure 2 (right) shows that M3ID continues to suffer from hallucinations associated with co-occurring object pairs (e.g., "person", "cup", "bottle") alongside the ground-truth object "dining table", which stem from training-time correlations. These observations together indicate that contrasting with a single hallucinated model, whether vision- or text-oriented, is insufficient to comprehensively address hallucinations across all stages of generation.

### 4.4 Remedy II: A Generalized form of Contrastive Decoding

The standard contrastive decoding formulation in equation 2 only allows the original outputs to be contrasted with one hallucinated model. However, as argued in section 4.3, a single hallucinated model may not be sufficient to capture different source of hallucination, and hence the resulting model after contrast will still suffer from unaddressed modes of hallucinations.

We propose a generalized form of contrastive decoding with multiple hallucinated models to contrast with. Our proposed generalized form takes the following form,

$$\log \boldsymbol{p}_t^{st} = (1 + \sum_{r=1}^{R} \alpha^{(r)}) \cdot \log \boldsymbol{p}_t^{orig} - \sum_{r=1}^{R} \alpha^{(r)} \cdot \log \boldsymbol{p}_t^{hal(r)}, \tag{9}$$

where $\alpha_r > 0 \ \forall r$. The generalized formulation empowers us to take advantage of multiple hallucinated models, representing different sources of hallucinations.

**Limitation & Assumption.** The participating hallucinated models $\boldsymbol{p}_t^{hal(r)}$ should be distinct enough to justify their inclusion. Adding similar hallucinated models under the generalized form will achieve the same result as the traditional contrastive decoding but with possibly higher latency.

## 5 CRoPS

Our proposal, CRoPS, integrates two hallucinated models under the generalized contrastive decoding formulation with the models being

$$
\begin{aligned}
\boldsymbol{p}_t^{\text{vis-hal}} &:= \text{softmax} \circ \text{LLM}_\theta(\overline{\boldsymbol{V}}, \boldsymbol{X}, \boldsymbol{y}_{<t}), \\
\boldsymbol{p}_t^{\text{vis-txt-hal}} &:= \text{softmax} \circ \text{LLM}_\theta(\overline{\boldsymbol{X}, \boldsymbol{y}_{<t}}).
\end{aligned}
\tag{10}
$$

The first model $\boldsymbol{p}_t^{\text{vis-hal}}$ is inspired by Huo et al. 2024 which removes important visual tokens (Section 3.3) to capture co-occurrence-related hallucinations (right subfigure of Figure 2) but *retains all text tokens*. The second model $\boldsymbol{p}_t^{\text{vis-txt-hal}}$ (as discussed in section 4.2) is deprived of all visual information and important textual information. During the early stages of generation, when generation is heavily dependent on visual tokens, $\boldsymbol{p}_t^{\text{vis-hal}}$ represents the source of hallucination whereas in the later stages when visual dependency diminishes, $\boldsymbol{p}_t^{\text{vis-txt-hal}}$ provides the hallucinated outputs by depriving the model of important text tokens. When combined, the resultant contrastive decoding, which we term as CRoPS can be written as equation 9 with the components defined as below,

$$
\begin{aligned}
\log \boldsymbol{p}_t^{\text{CRoPS}} := \left(1 + \alpha^{(1)} + \alpha_t^{(2)}\right) \cdot \log \boldsymbol{p}_t^{\text{orig}} \\
- \alpha^{(1)} \cdot \log \boldsymbol{p}_t^{\text{vis-hal}} - \alpha_t^{(2)} \cdot \log \boldsymbol{p}_t^{\text{vis-txt-hal}}
\end{aligned}
\tag{11}
$$

where $\alpha^{(1)} \equiv \alpha$ and $\alpha_t^{(2)} := (1 - e^{-\gamma t})/e^{-\gamma t}$. Since vision-driven hallucinations occur predominantly in the early stages of generation, $\alpha^{(1)}$ is kept constant. In contrast, $\alpha_t^{(2)}$ is designed to gradually increase over time, reflecting the growing influence of text-induced hallucinations in the later stages. Similar to prior works, we add *confidence* and *plausibility* constraint while contrasting via equation 11. The overall algorithm is outlined in the Appendix.

**Distinctiveness of $\boldsymbol{p}_t^{\text{vis-hal}}$ and $\boldsymbol{p}_t^{\text{vis-txt-hal}}$.** The generalized contrastive decoding with multiple hallucinated models is only justified by how distinctive the participating models are. Left subplot of Figure 4 shows Jensen-Shannon Divergence (JSD) between our proposed $\boldsymbol{p}_t^{\text{vis-hal}}$ and $\boldsymbol{p}_t^{\text{vis-txt-hal}}$. It is compared with the maximum ($\log 2$), and the corresponding JSD between hallucinated models used by previous works, indicating that simply stacking the hallucinated models from previous works under generalized contrastive is sub-optimal.

**Visual Dependency of Final Outputs.** The right subplot of Figure 4 shows the VD(t) of the final outputs of competing methods and $\boldsymbol{p}_t^{\text{CRoPS}}$. It is clear that CRoPS is more visually grounded (i.e., less hallucinated) than vanilla sampling and other methods.

## 6 Experiments

### 6.1 Experimental Settings

**Models and Baselines.** We conduct evaluations on three widely adopted LVLMs: LLaVA-1.5 (Liu et al., 2023b), LLaVA-NeXT (Liu et al., 2024a), and Qwen2-VL (Wang et al., 2024a). For baseline comparisons, we consider several recent training-free hallucination mitigation techniques. These include VCD (Leng et al., 2024), ICD (Wang et al., 2024b), OPERA (Huang et al., 2024), ClearSight (Yin et al., 2025), SID (Huo et al., 2024) and M3ID (Favero et al., 2024). These methods represent state-of-the-art strategies that aim to reduce hallucinations without requiring additional fine-tuning or retraining of the underlying vision-language models.

**Implementation Details.** We set the pruning layer to $l = 2$ for both text and visual tokens. We follow (Huo et al., 2024) choice, as they demonstrate that attention scores in the initial layers better reflect token importance and are relatively unaffected by attention sinks. Our experiments utilized the 7B and 13B backbones of LLaVA-1.5, 7B backbone of Qwen2-VL, and 8B backbone of LLaVA-NeXT. We applied nucleus sampling with a top-p value of 0.9 and a temperature of 1. All experiments were conducted using three

Table 1: Evaluation of vision–language grounding on the MS-COCO validation set (Lin et al., 2014). Captions are generated using the prompt *"Please describe this image in detail"*. CHAIR scores $C_S$ and $C_I$ represent the percentages of hallucinated objects and captions, respectively, where lower values indicate stronger visual grounding. Recall denotes the percentage of annotated objects correctly mentioned in the generated captions. All results are averaged over three random seeds.

| Method | LLaVA-1.5 (7B) | | | LLaVA-1.5 (13B) | | | LLaVA-NeXT | | | Qwen2-VL | | |
|---|---|---|---|---|---|---|---|---|---|---|---|---|
| | $C_S \downarrow$ | $C_I \downarrow$ | Recall $\uparrow$ | $C_S \downarrow$ | $C_I \downarrow$ | Recall $\uparrow$ | $C_S \downarrow$ | $C_I \downarrow$ | Recall $\uparrow$ | $C_S \downarrow$ | $C_I \downarrow$ | Recall $\uparrow$ |
| Sampling | 57.0 | 17.0 | 75.0 | 50.2 | 13.7 | 76.4 | 37.4 | 8.9 | 66.3 | 33.2 | 8.0 | 68.1 |
| ClearSight | 54.1 | 16.2 | 74.3 | 49.4 | 13.9 | 74.8 | 35 | 8.5 | 63.6 | 13.5 | 8.7 | 38.2 |
| VCD | 53.3 | 15.3 | 77.9 | 49.5 | 13.7 | 77.7 | 36.4 | 8.8 | 68.6 | 29.0 | 7.8 | 67.3 |
| ICD | 52.5 | 14.6 | 77.7 | 49.2 | 13.9 | 78.1 | 36.6 | 9.4 | 67.1 | 28.0 | 7.9 | 66.1 |
| OPERA | 49.1 | 13.8 | 78.5 | 48.2 | 13.2 | 78.9 | 35.5 | 8.9 | 66.9 | 31.0 | 8.1 | 67.9 |
| SID | 48.9 | 13.0 | 77.9 | 47.0 | 12.3 | 77.9 | 37.0 | 10.7 | 70.2 | 30.6 | 8.1 | 66.1 |
| M3ID | 47.1 | 12.8 | 74.8 | 45.5 | 12.2 | 75.3 | 36.1 | 9.8 | 68.7 | 28.8 | 7.3 | 64.8 |
| *CRoPS* | **39.5** | **10.2** | 76.3 | **38.5** | **9.1** | 75.1 | **33.2** | **8.1** | 66.2 | **26.9** | **6.9** | 67.4 |

Table 2: Performance on the AMBER benchmark (Wang et al., 2023a), evaluated with the prompt *"Please describe this image in detail"*. We report three axes of hallucination: **HAL** (overall hallucination rate), **Cog** (cognitive deviation from correct attributes/relations), and **CHAIR** (object-level hallucination), where lower is better. Results are averaged over three random seeds.

| Method | LLaVA-1.5 (7B) | | | LLaVA-1.5 (13B) | | | LLaVA-NeXT | | | Qwen2-VL | | |
|---|---|---|---|---|---|---|---|---|---|---|---|---|
| | CHAIR $\downarrow$ | Hal $\downarrow$ | Cog $\downarrow$ | CHAIR $\downarrow$ | Hal $\downarrow$ | Cog $\downarrow$ | CHAIR $\downarrow$ | Hal $\downarrow$ | Cog $\downarrow$ | CHAIR $\downarrow$ | Hal $\downarrow$ | Cog $\downarrow$ |
| Sampling | 10.6 | 44.3 | 4.0 | 9.3 | 41.3 | 4.2 | 10.5 | 57.1 | 4.1 | 6.4 | 38.9 | 2.8 |
| ClearSight | 10.5 | 44.1 | 4.2 | 9.2 | 40.5 | 4.0 | 9.1 | 54.4 | 4.5 | 5.9 | 34.0 | 2.2 |
| VCD | 9.0 | 42.9 | 4.6 | 8.4 | 38.3 | 3.9 | 9.5 | 55.7 | 4.2 | 5.7 | 33.9 | 2.4 |
| ICD | 10.0 | 44.8 | 4.3 | 8.5 | 39.5 | 4.1 | 10.1 | 53.0 | 4.5 | 6.0 | 34.1 | 2.5 |
| OPERA | 9.8 | 43.0 | 4.5 | 8.7 | 40.0 | 4.1 | 9.8 | 51.0 | 4.3 | 6.3 | 35.0 | 2.6 |
| SID | 9.3 | 43.7 | 3.7 | 6.9 | 35.0 | 3.5 | 9.1 | 54.2 | 3.9 | 5.4 | 30.6 | 1.8 |
| M3ID | 9.0 | 40.0 | 3.0 | 7.9 | 40.0 | 2.9 | 8.7 | 51.9 | 3.1 | 5.5 | 27.9 | 1.5 |
| *CRoPS* | **6.3** | **29.3** | **2.8** | **5.7** | **27.8** | **2.5** | **7.2** | **44.6** | **2.6** | **5.1** | **24.2** | **1.1** |

random seeds, and the average performance across these runs is reported. The hyperparameter configurations and their ablation analysis are provided in Section 7.

## 6.2 Experimental Results

In this section, we evaluate **CRoPS** across a range of benchmarks that capture different forms of hallucination and multimodal reasoning. For each benchmark, we briefly describe what it measures and then discuss the corresponding results.

**CHAIR Benchmark.** The CHAIR (Captioning Hallucination Assessment with Image Relevance) benchmark (Rohrbach et al., 2018) evaluates object-level hallucinations by comparing nouns in generated captions with ground-truth object annotations, where lower $C_S$ and $C_I$ indicate stronger visual grounding. As shown in Table 1, **CRoPS** achieves the lowest CHAIR scores across all LVLMs, outperforming prior training-free methods by a clear margin. On the LLaVA-1.5 series, CRoPS reduces hallucination rates by roughly **15–25%** relative to M3ID, while maintaining comparable recall. The trend continues for LLaVA-NeXT and Qwen2-VL, where CRoPS yields an additional **8–10%** reduction in $C_S$ and $C_I$ without any degradation in descriptive quality. Overall, these consistent percentage gains across architectures highlight that CRoPS effectively mitigates hallucinations without relying on retraining or auxiliary supervision. Although Clear-Sight attains a lower $C_S$ on Qwen2-VL, this improvement comes at the expense of recall (38.2 vs. 67.4), suggesting over-suppression of visual details. In contrast, CRoPS preserves a balanced trade-off between grounding accuracy and linguistic richness, producing visually faithful yet detailed image descriptions.

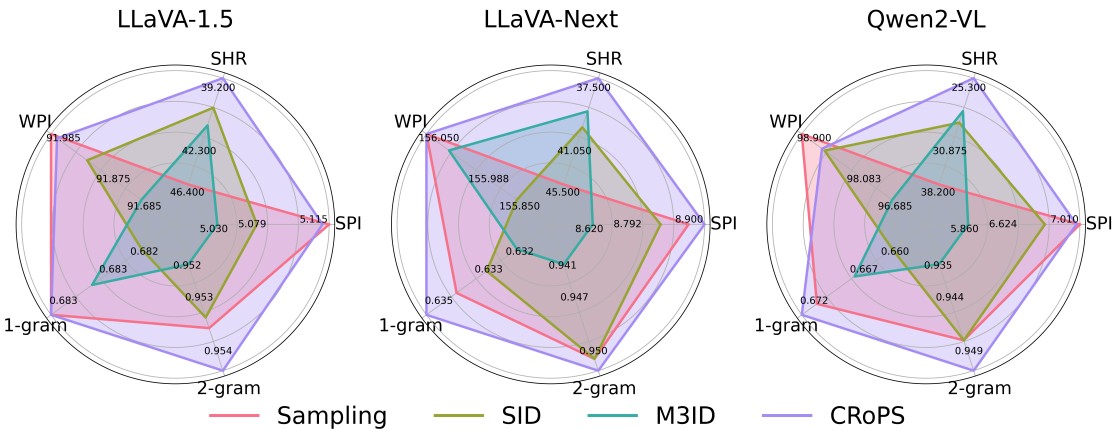

Figure 5: Evaluation on the GPT-4o assisted benchmark (Zhou et al., 2024), comparing hallucination (SHR), fluency (1- and 2-gram precision), and descriptive detail (WPI and SPI). Larger enclosed areas correspond to better overall performance. Please zoom in for clearer visualization.

Table 3: Evaluation results on the POPE VQA hallucination benchmark (Li et al., 2023). POPE comprises three subsets: *Random*, *Popular*, and *Adversarial*. Each sample follows the query template: "Is a *object* present in the image?", where *object* is selected either randomly (Random), from the most frequent dataset objects (Popular), or from objects that frequently co-occur with the target entity (Adversarial). We report the average performance across all three subsets; detailed results are provided in the Appendix G.

| Method | LLaVA-1.5 | | LLaVA-NeXT | | Qwen2-VL | |
|---|---|---|---|---|---|---|
| | Acc. ↑ | F1 ↑ | Acc. ↑ | F1 ↑ | Acc. ↑ | F1 ↑ |
| Sampling | 81.7 | 82.8 | 87.3 | 87.6 | 84.2 | 82.8 |
| ClearSight | 82.0 | 83.0 | 87.3 | 87.7 | 73.5 | 74.5 |
| VCD | 82.6 | 83.3 | 87.7 | 88.1 | 84.7 | 83.2 |
| ICD | 82.2 | 82.9 | 86.6 | 87.4 | 85.2 | 84.3 |
| OPERA | 82.3 | 82.9 | 87.4 | 88.2 | 85.5 | 84.5 |
| SID | 82.7 | 83.3 | 88.6 | 88.4 | 85.6 | 83.9 |
| M3ID | 82.4 | 83.4 | 88.0 | 88.1 | 84.9 | 83.5 |
| *CRoPS* | **83.9** | **84.6** | **89.4** | **89.4** | **86.1** | **85.3** |

**AMBER Benchmark.** This is a multi-dimensional hallucination benchmark that evaluates not just whether objects are wrongly included (via CHAIR), but also how the model misrepresents attributes or relations (HAL) and deviates in cognitive consistency (Cog) in generated captions (Wang et al., 2023a). Here, **HAL** captures the overall hallucination frequency, and **Cog** measures the divergence between described attributes or relations and ground truth. As shown in Table 2, **CRoPS** consistently outperforms all baselines across models. On the LLaVA-1.5 series, CRoPS achieves about a **25–30% reduction** in both CHAIR and HAL, and a **10–15% reduction** in Cog, demonstrating more accurate and semantically coherent descriptions. For LLaVA-NeXT and Qwen2-VL, it maintains similar improvements, reducing hallucination metrics by roughly **20–25%** while preserving fluency and descriptive richness. These consistent percentage reductions across architectures highlight that CRoPS mitigates both perceptual and cognitive hallucinations more effectively than prior training-free methods.

**GPT-4o Assisted Benchmark.** The GPT-4o assisted benchmark (Zhou et al., 2024) uses Visual Genome annotations as ground truth and prompts GPT-4o to evaluate hallucination at a finer granularity. It measures the Sentence-Level Hallucination Ratio (SHR), indicating how often generated sentences contain hallucinated content, and also evaluates fluency (1-gram and 2-gram precision) and descriptiveness (Words Per Image, WPI). As shown in Figure 5, **CRoPS** achieves the lowest SHR across all model variants, demonstrating its

Table 4: Results on the General-Purpose benchmark showing **MME** and **MathVista** scores for LLaVA-1.5, LLaVA-NeXT, and Qwen2-VL. Higher values indicate better performance. Bold numbers mark the best result and underlined numbers mark the second-best within each column. ***CRoPS*** achieves the top scores across most settings.

| Method | LLaVA-1.5 | | LLaVA-NeXT | | Qwen2-VL | |
|---|---|---|---|---|---|---|
| | MME ↑ | MathVista ↑ | MME ↑ | MathVista ↑ | MME ↑ | MathVista ↑ |
| Sampling | 1601 | 27.4 | 1669 | 34.8 | 2058 | **56.9** |
| ClearSight | 1569 | 27.1 | 1602 | 35.1 | 2001 | 54.7 |
| VCD | 1622 | 26.5 | 1630 | 36.2 | 2070 | 55.1 |
| ICD | 1605 | 25.9 | 1664 | 35.0 | 2060 | 54.2 |
| OPERA | 1615 | 27.0 | 1650 | 35.5 | 2080 | 54.8 |
| SID | 1634 | 26.3 | 1607 | 36.8 | 2097 | 54.3 |
| M3ID | 1607 | 26.9 | 1683 | 36.6 | 2090 | 52.0 |
| ***CRoPS*** | **1662** | **28.9** | **1779** | **38.0** | **2184** | 55.6 |

strength in suppressing relational and attribute-level hallucinations. Moreover, CRoPS maintains superior 1- and 2-gram fluency and higher WPI, indicating fluent and detailed captions. Unlike other baselines that shorten output to reduce hallucinations, CRoPS preserves text richness while improving factual grounding.
**POPE.** The POPE benchmark (Li et al., 2023) evaluates hallucination detection in a binary visual question answering format, where the model must confirm or deny the presence of an object in the image. As shown in Table 3, **CRoPS** achieves the highest Accuracy and F1 across all LVLMs, showing an average performance gain of about **2%** over the strongest baseline. This consistent improvement demonstrates that CRoPS remains robust even under restricted yes/no answer formats, maintaining strong visual grounding across models.

While M3ID performs competitively, it slightly lags behind SID and CRoPS in this binary setting. This difference arises because POPE differs from captioning benchmarks, it requires only concise yes/no responses, which limit the degree of visual token dilution typically observed in open-ended generation. However, as discussed in M3ID (Favero et al., 2024), a non-negligible visual token dilution effect persists due to the structural separation between image tokens and output tokens introduced by the VQA prompt template. To account for this offset, we follow the same configuration and select the decoding position $t = t_0$, where $t_0$ corresponds to the number of tokens between the image and the answer span. This adjustment ensures that CRoPS aligns visual grounding signals with the output region, resulting in consistently higher precision and balanced grounding across all models. Detailed results are provided in Appendix G.

**MME and MathVista Evaluations.** The MME benchmark (Liang et al., 2024) evaluates general multimodal perception and recognition. Table 4 shows that CRoPS improves MME scores across architectures, indicating stronger grounding in diverse perceptual tasks. MathVista (Lu et al., 2024) probes visual mathematical reasoning; CRoPS achieves top or near-top performance across models, suggesting its decoding strategy benefits structured multimodal reasoning beyond captioning. These results highlight that CRoPS's contrastive design enhances both descriptive fidelity and logical reasoning across diverse LVLM tasks.

# 7 Analysis

**Latency Comparison.** Table 5 shows inference time and peak GPU memory on LLaVA-1.5 7B and the CHAIR benchmark. CRoPS adds minimal overhead despite an extra forward pass due to its lightweight input design. Performance gains stem from the targeted contrastive design rather than repeated model calls. Compared to 5-beam search, CRoPS achieves lower hallucination with only a modest increase in time and memory. This analysis uses a vanilla implementation; an optimized version would further reduce overhead.

Table 5: Efficiency Comparison on NVIDIA A100. Time is measured in seconds and memory usage MB.

| Method | Time ↓ | Memory ↓ | $C_S$ ↓ |
|---|---|---|---|
| Sampling | 215 | 15699 | 57 |
| Beam Search (5 beams) | 531 | 16737 | 50.7 |
| VCD | 550 | 17864 | 53.3 |
| SID | 510 | 16574 | 48.93 |
| OPERA | 1947 | 21943 | 49.1 |
| CRoPS | 652 | 16934 | **39.46** |

Table 6: **Left:** Effect of image and text removal on the hallucinated model. **Right:** Comparison of token retention policies on CHAIR metrics and Recall.

| Image Removal | Text Removal | $C_S \downarrow$ | $C_I \downarrow$ | Recall $\uparrow$ | Policy | $C_S \downarrow$ | $C_I \downarrow$ | Recall $\uparrow$ |
|---|---|---|---|---|---|---|---|---|
| Full | No | 45.4 | 12.4 | 73.3 | Constant | 43.6 | 10.9 | 75.0 |
| No | Partial (Least Important) | 47.8 | 13.2 | 73.8 | All-but-one | 46.8 | 13.1 | 73.6 |
| Full | Partial (Least Important) | **43.2** | **10.8** | 73.3 | Linear | 41.2 | 10.7 | 74.6 |
| Full | Partial (Random) | 50.7 | 13.8 | 76.4 | Ours | **39.5** | **10.2** | **76.3** |

**Effect of Image and Text Removal.** Left sub-table of Table 6 analyzes the impact of image and text removal on hallucination behavior. *Row 1* removes all image tokens while keeping text tokens. *Row 2* retains all image tokens but removes the least important text tokens. *Row 3* is our proposed setting, removing all image tokens while partially preserving text tokens. *Row 4* removes the same number of text tokens but at random. Our method outperforms others, confirming that selective removal of the least important text tokens is more effective (see section 4.2).

**Choice of Hallucination Model.** To test our contrastive hallucination mitigation strategy, we replace the second hallucinated model with M3ID while keeping CRoPS unchanged. Table 7 shows this yields inferior results, underscoring the effectiveness of our proposed second model design.

Table 7: Ablation on CHAIR metric on MS-COCO dataset on the choice of hallucination model.

| Method | $C_S \downarrow$ | $C_I \downarrow$ | Recall $\uparrow$ |
|---|---|---|---|
| M3ID + SID | 45.1 | 11.8 | 78.3 |
| **CRoPS** | **39.5** | **10.2** | 76.3 |

**Token Retention Policy.** We control the number of retained tokens at each generation step $t$ using a policy $\eta(t)$. The baselines include: **Constant** ($\eta(t) = \beta_0$), which keeps a fixed number $\beta_0$ of tokens; **All-but-one** ($\eta(t) = t - 2$), which discards all tokens except the most recent one; and **Linear** ($\eta(t) = \beta_0 + \beta_1 t$), which ensures a steady linear increase in the number of retained tokens as $t$ grows. To enable a smooth yet saturating increase, we introduce an **exponential policy** defined as $\eta(\mu, t) = \beta_0 + \beta_1(1 - e^{-\mu t})$, where $\mu > 0$ controls the rate of growth. The policy starts at $\beta_0$ and asymptotically approaches $\beta_0 + \beta_1$, thereby preventing abrupt context drops and yielding empirically more stable generation. As shown in right sub-table of Table 6, our exponential policy achieves the lowest hallucination scores, outperforming all baseline strategies.

**Hyperparameter Ablation.** We analyze the sensitivity of CRoPS to its key hyperparameters in Eq. 11. The coefficients $\alpha^{(1)}$ and $\alpha_t^{(2)}$ control the overall strength and the time-dependent growth of the contrastive penalties, respectively. The parameters $\beta_0$, $\beta_1$, and $\mu$ govern the dynamic weighting function $\eta$, which determines how many text tokens are masked at each step. Here, $\beta_0$ and $\beta_1$ define the lower and upper bounds of the retained-token range, while $\mu$ controls how smoothly $\eta$ increases over time. As shown in Table 8, moderate contrastive strength and gradual masking yield the best re-

Table 8: Effect of varying hyper-parameters on CRoPS performance.

| $\alpha$ | $\gamma$ | $b_0$ | $b_1$ | $\mu$ | $C_S$ |
|---|---|---|---|---|---|
| 1.0 | 0.02 | 10 | 30 | $1 \times 10^{-3}$ | 39.5 |
| 0.5 | 0.05 | 5 | 20 | $1 \times 10^{-3}$ | 42.3 |
| 1.0 | 0.02 | 5 | 40 | $1 \times 10^{-3}$ | 41.2 |
| 1.5 | 0.01 | 10 | 30 | $1 \times 10^{-2}$ | 43.5 |
| 1.0 | 0.02 | 10 | 40 | $1 \times 10^{-3}$ | 40.7 |

sults, whereas extreme values of $\alpha$ or $\mu$ degrade performance by either weakening or over-suppressing the contrastive signal.

For all experiments, we adopt a single shared configuration of hyperparameters: $\alpha = 1.0$, $\gamma = 0.02$, $\beta_0 = 10$, $\beta_1 = 30$, and $\mu = 1e-3$. This configuration is used consistently across all backbones and all benchmarks, and no model-specific or dataset-specific tuning is employed.

## 8 Conclusion and Limitations

In this work, we first highlighted how existing mitigation techniques, which address visual token dilution and attention to irrelevant visual features individually, fail to comprehensively tackle both. To bridge this

gap, we introduced CRoPS, a novel decoding strategy that effectively mitigates both sources of hallucination while incurring minimal additional computational overhead, and demonstrates significant gains on multiple benchmarks.

**Limitations.** While our study provides valuable insights into the weaknesses of contrastive decoding, it is not exhaustive. We specifically focus on certain types of hallucinations, leaving the exploration to future work. Additionally, our analysis is restricted to the latest SOTA methods and only considers training-free contrastive decoding approaches. Exploring alternative frameworks, including those that involve task-specific training, could provide a more complete understanding of the trade-offs in contrastive decoding.

Our approach also introduces some additional inference latency. Although CRoPS is about $3\times$ slower than vanilla sampling (652s vs. 215s), its runtime remains comparable to or lower than existing training-free methods such as VCD (550s), SID (510s) and OPERA (1947s). The overhead comes from an extra forward pass, but this pass is relatively lightweight since the text-deficit model processes only a small subset of tokens. Latency remains a practical limitation. Future work can reduce this overhead, for example by parallelizing the forward passes required for generating the hallucinated models.

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

## A    Multi-Modal Mutual-Information Decoding (M3ID)

M3ID proposes to utilize the same LLM model to generate the hallucinated probabilities by completely removing the visual tokens $\boldsymbol{V}$ from the input. Formally, M3ID defines,

$$
\begin{aligned}
\boldsymbol{p}_t^{orig} &:= \mathrm{softmax} \circ \mathrm{LLM}_\theta\big(\boldsymbol{V}, \boldsymbol{X}, \boldsymbol{y}_{<t}\big), \\
\boldsymbol{p}_t^{hal} &:= \mathrm{softmax} \circ \mathrm{LLM}_\theta\big(\boldsymbol{X}, \boldsymbol{y}_{<t}\big), \\
\alpha_t &:= (1 - e^{-\gamma t})/e^{-\gamma t}, \\
\log \boldsymbol{p}_t^{\mathrm{M3ID}} &:= \log \boldsymbol{p}_t^{or} + \alpha_t \cdot (\log \boldsymbol{p}_t^{orig} - \log \boldsymbol{p}_t^{hal}).
\end{aligned}
\tag{12}
$$

Notice how the mixing coefficient $\alpha$ is time-dependent, and is a monotonically increasing function of inference time step $t$. M3ID justifies this by arguing that the hallucination gets stronger as $t$ increases, or equivalently, the effect of $\boldsymbol{p}_t^{hal}$ gets stronger.

## B    Self Introspective Decoding (SID).

SID selectively retains visual tokens with low coherency rather than removing all the visual tokens from the input of the LLM. The coherency of visual tokens is determined via a summary attention score assigned to every visual input token. The score $\psi(\boldsymbol{y}_t)$ for each visual token $\boldsymbol{v}_i$ is then computed as,

$$
\psi(\boldsymbol{y}_t) = \frac{1}{H} \sum_{h=1}^{H} \mathrm{Attention}^{(h)}\big(\boldsymbol{v}_i, \boldsymbol{y}_t\big),
\tag{13}
$$

Using this score, SID chooses the bottom $\bar{m}, (< m = |\boldsymbol{V}|)$ visual tokens with the lowest $\psi$ scores and removes high-importance tokens to create a new sparser set of visual tokens $\overline{\boldsymbol{V}} := \mathrm{Sparse}\big(\boldsymbol{V}\big) \mathrm{with} |\overline{\boldsymbol{V}}| = \bar{m}$.

Considering that modern architectures like LLaVA-Next and Qwen2-VL do not have a fixed number of image tokens, we set $\bar{m}$ to be 25% of the total number of image tokens dynamically. Finally, the probability outputs from the weak model are obtained as

$$
\begin{aligned}
\boldsymbol{p}_t^{orig} &:= \mathrm{softmax} \circ \mathrm{LLM}\theta\big(\boldsymbol{V}, \boldsymbol{X}, \boldsymbol{y}_{<t}\big), \\
\boldsymbol{p}_t^{hal} &:= \mathrm{softmax} \circ \mathrm{LLM}\theta\big(\overline{\boldsymbol{V}}, \boldsymbol{X}, \boldsymbol{y}_{<t}\big), \\
\log \boldsymbol{p}_t^{\mathrm{SID}} &:= (1 + \alpha) \cdot \log \boldsymbol{p}_t^{orig} - \alpha \cdot \log \boldsymbol{p}_t^{hal}.
\end{aligned}
\tag{14}
$$

The mixing coefficient $\alpha$ from equation 2 is a hyper-parameter in SID, and unlike time-dependent mixing in M3ID, SID has a constant $\alpha$ throughout the inference time steps.

## C    Detailed Comparison of M3ID vs. Our Novel Hallucinated Model

| Method | LLaVA-1.5 | | | LLaVA-NeXT | | | Qwen2-VL | | |
|---|---|---|---|---|---|---|---|---|---|
| | $C_S \downarrow$ | $C_I \downarrow$ | Recall $\uparrow$ | $C_S \downarrow$ | $C_I \downarrow$ | Recall $\uparrow$ | $C_S \downarrow$ | $C_I \downarrow$ | Recall $\uparrow$ |
| M3ID | 45.4 | 12.4 | 73.3 | 36.1 | 9.8 | 68.7 | 28.8 | 7.3 | 64.8 |
| Novel Model $\boldsymbol{p}^{\mathrm{vis\text{-}txt\text{-}hal}}$ | 43.2 | 10.8 | 73.3 | 35.2 | 9.4 | 66.2 | 28.0 | 7.4 | 66.6 |

Table 9: Ablation results comparing M3ID and the Novel Hallucinated Model $\boldsymbol{p}^{\mathrm{vis\text{-}txt\text{-}hal}}$ across different models using CHAIR metrics.

Table 9 shows the performance gain of the proposed hallucinated model $\boldsymbol{p}^{\mathrm{vis\text{-}txt\text{-}hal}}$ from section section 5, which selects the least important text tokens instead of all text tokens, as in M3ID. By selecting only the least important tokens, following the same approach as SID (vision token selection), we observed a performance improvement on the CHAIR benchmark across all models.

---

**Algorithm 1** *CRoPS*

---

**Require:** LLM $\text{LLM}_\theta$, Text $\boldsymbol{X} = (\boldsymbol{x}_1, \ldots, \boldsymbol{x}_n)$, Image $\boldsymbol{V} = (\boldsymbol{v}_1, \ldots, \boldsymbol{v}_m)$
**Require:** Hyperparams: $\eta(\cdot, \cdot), \bar{m}, \gamma, \alpha$
**Ensure:** Output sequence $\boldsymbol{Y} = (\boldsymbol{y}_1, \ldots, \boldsymbol{y}_T)$

1: $\boldsymbol{y}_0 \leftarrow \text{BOS}, t \leftarrow 1, \boldsymbol{Y}_{<1} \leftarrow (\boldsymbol{y}_0)$
2: **while** $\boldsymbol{y}_{t-1} \neq \text{EOS}$ **do**
3: $\quad \overline{\boldsymbol{V}} \leftarrow \text{LeastImp}[\boldsymbol{V}, \bar{m}]$
4: $\quad \overline{\boldsymbol{X}, \boldsymbol{y}_{<t}} \leftarrow \text{LeastImp}[(\boldsymbol{X}, \boldsymbol{y}_{<t}), n + t - \eta(n, t)]$
5: $\quad \boldsymbol{p}_t^{orig} \leftarrow \text{softmax} \circ \text{LLM}_\theta(\boldsymbol{V}, \boldsymbol{X}, \boldsymbol{y}_{<t})$
6: $\quad \boldsymbol{p}_t^{\text{vis-hal}} \leftarrow \text{softmax} \circ \text{LLM}_\theta(\overline{\boldsymbol{V}}, \boldsymbol{X}, \boldsymbol{y}_{<t})$
7: $\quad \boldsymbol{p}_t^{\text{vis-txt-hal}} \leftarrow \text{softmax} \circ \text{LLM}_\theta(\overline{\boldsymbol{X}, \boldsymbol{y}_{<t}})$
8: $\quad \alpha_t^{(1)} \leftarrow \frac{1 - e^{-\gamma t}}{e^{-\gamma t}}, \alpha_t^{(2)} \leftarrow \alpha$
9: $\quad \log \boldsymbol{p}_t^{\text{CRoPS}} \leftarrow \left(1 + \alpha_t^{(1)} + \alpha_t^{(2)}\right) \cdot \log \boldsymbol{p}_t^{orig}$
$\qquad - \alpha_t^{(1)} \cdot \log \boldsymbol{p}_t^{\text{vis-hal}} - \alpha_t^{(2)} \cdot \log \boldsymbol{p}_t^{\text{vis-txt-hal}}$
10: $\quad \boldsymbol{y}_t \leftarrow \text{Decode}(\boldsymbol{p}_t^{\text{CRoPS}})$
11: $\quad \boldsymbol{Y}_{<t+1} \leftarrow (\boldsymbol{y}_{<t} : \boldsymbol{y}_t)$
12: $\quad t \leftarrow t + 1$
13: **end while**

---

# D  Benchmarks and Evaluation Metrics

LVLM hallucination benchmarks can be broadly categorized into generative and discriminative approaches. **Generative benchmarks** evaluate hallucination in free-form text generation. In our evaluation of CRoPS we use 1) CHAIR (Rohrbach et al., 2018): This metric measures hallucination by comparing the objects mentioned in generated captions with the annotated objects present in the corresponding image. 2) AMBER (Wang et al., 2023a): AMBER quantifies the proportion of hallucinated responses (Hal) and assesses their alignment with human cognition (Cog). 3) GPT-4-assisted benchmarks (Zhao et al., 2023): These benchmarks utilize fine-grained, object-level descriptions from the Visual Genome dataset (Krishna et al., 2017) and employ GPT-4 to calculate the Sentence-level Hallucination Ratio (SHR). Additionally, we compute n-gram fluency (with $n = 1, 2$) to assess text smoothness, and we analyze verbosity and detail by measuring Words Per Image (WPI) and Sentences Per Image (SPI). **Discriminative benchmarks** evaluate hallucination in a Visual Question Answering setting, where responses are typically binary (e.g., "yes" or "no"), making evaluation similar to a classification task. We use POPE (Li et al., 2023), which frames object hallucination as a binary classification problem with questions of the form "Is a $\langle$object$\rangle$present in the image?"

Beyond hallucination-specific evaluations, we assess the general capabilities of LVLMs using 1) MME (Liang et al., 2024): A comprehensive benchmark comprising ten sub-tasks that assess perceptual capabilities, as well as four sub-tasks that evaluate recognitive abilities via yes/no questions. and 2) MathVista (Lu et al., 2024): A benchmark designed to analyze mathematical reasoning capabilities in visually complex scenarios.

# E  Informativeness of Past Generated Tokens

From equation 12,

$$\boldsymbol{p}_t^{orig} := \text{softmax} \circ \text{LLM}_\theta(\boldsymbol{V}, \boldsymbol{X}, \boldsymbol{y}_{<t}),$$

$$\boldsymbol{p}_t^{hal} := \text{softmax} \circ \text{LLM}_\theta(\boldsymbol{X}, \boldsymbol{y}_{<t}),$$

$$\log \boldsymbol{p}_t^{\text{M3ID}} := \log \boldsymbol{p}_t^{orig} + \alpha_t \cdot (\log \boldsymbol{p}_t^{orig} - \log \boldsymbol{p}_t^{hal})$$

where, $\alpha_t := (1 - e^{-\gamma t})/e^{-\gamma t}$

And, from equation 4, Visual Dependency (VD) is defined as,

$$\text{VD}(t) := \text{dist}\Big(\text{softmax} \circ \text{LLM}_\theta(\boldsymbol{V}, \boldsymbol{X}, \boldsymbol{y}_{<t}), \ \text{softmax} \circ \text{LLM}_\theta(\boldsymbol{X}, \boldsymbol{y}_{<t})\Big)$$

Now, as $\mathrm{VD}(t) \to 0$,

$$\mathrm{softmax} \circ \mathrm{LLM}_\theta\left(\boldsymbol{X}, \boldsymbol{y}_{<t}\right) \to \mathrm{softmax} \circ \mathrm{LLM}_\theta\left(\boldsymbol{V}, \boldsymbol{X}, \boldsymbol{y}_{<t}\right)$$

$$\implies \boldsymbol{p}_t^{hal} \to \boldsymbol{p}_t^{orig}$$

Hence,

$$\lim_{\mathrm{VD}(t) \to 0} \mathbf{p}_t^{\mathrm{M3ID}} = \mathbf{p}_t^{orig}.$$

## F   Ablation on Contrastive Ordering Strategies

To assess the effectiveness of our proposed contrastive hallucination mitigation strategy, we perform an ablation study comparing different contrastive application orders across weak models in the **LLaVA-1.5-7B** setting. Specifically, we compare three variants:

- **SID → M3ID**: Applying contrastive Decoding using hallucinated model of SID, followed by contrasting using hallucinated model of M3ID.

- **M3ID → SID**: Applying contrastive Decoding using hallucinated model of M3ID, followed by contrasting using hallucinated model of SID.

- **Novel Weak Model (Ours)**: Applying contrastive decoding with a modified hallucinated model of M3ID , followed by contrasting with hallucinated model of SID

The results are summarized below:

Table 10: Ablation CHAIR metric on MSCOCO dataset on the ordering of contrastive strategies. Lower CHAIR metrics indicate fewer hallucinations, while higher Recall indicates answer fidelity.

| Method | CHAIRs ↓ | CHAIRi ↓ | Recall ↑ |
|---|---|---|---|
| SID → M3ID | 45.5 | 11.9 | 78.9 |
| M3ID → SID | 45.1 | 11.8 | 78.3 |
| **Novel Weak Model (Ours)** | **39.5** | **10.2** | 76.3 |

The **CHAIRs** and **CHAIRi** metrics,which directly quantify hallucination improve significantly, suggesting that our novel hallucinated model is better at rejecting hallucinated content.

This ablation highlights a key insight: *naïvely composing existing contrastive decoding is insufficient.* Simply applying M3ID and SID in different orders improves hallucination scores but it can be further enhanced via our novel hallucinated model.

## G   Detailed Results on POPE Benchmark

Table 11 reports the detailed results on the POPE benchmark across its three subsets: *Random*, *Popular*, and *Adversarial*. CRoPS consistently outperforms all baselines across all model variants and subsets in both Accuracy and F1. The gains are most pronounced on the Adversarial subset, where models are challenged with highly co-occurring object pairs, highlighting CRoPS's ability to remain visually grounded even in confusing visual contexts. Across all subsets, the improvement over M3ID and SID averages around 1–2% in both metrics, demonstrating that CRoPS generalizes well across different levels of difficulty. Unlike attention-adjustment methods such as ClearSight, which degrade sharply on Qwen2-VL, CRoPS maintains balanced and robust performance across architectures. These results reinforce that CRoPS effectively mitigates hallucinations in binary grounding tasks while preserving response accuracy.

Table 11: Results on the MS-COCO split of the POPE benchmark

| Dataset Type | Method | LLaVA-1.5 (7B) | | LLaVA-NeXT | | Qwen2-VL | |
|---|---|---|---|---|---|---|---|
| | | Acc. ↑ | F1 ↑ | Acc. ↑ | F1 ↑ | Acc. ↑ | F1 ↑ |
| Random | Sampling | 85.7 | 86.0 | 91.0 | 90.8 | 86.1 | 84.5 |
| | SID | 87.1 | 86.8 | 91.4 | 91.5 | 87.7 | 84.9 |
| | M3ID | 86.8 | 87.0 | 91.0 | 91.4 | 86.4 | 85.7 |
| | ClearSight | 86.8 | 86.5 | 90.7 | 90.9 | 71.5 | 70.8 |
| | VCD | 86.5 | 86.2 | 91.1 | 91.2 | 86.2 | 85.0 |
| | ICD | 86.1 | 86.3 | 89.7 | 89.5 | 86.7 | 86.0 |
| | OPERA | 86.3 | 86.6 | 89.9 | 90.5 | 86.8 | 86.2 |
| | *CRoPS* | **87.8** | **87.7** | **92.0** | **92.3** | **87.9** | **86.8** |
| Popular | Sampling | 82.8 | 83.6 | 88.1 | 88.2 | 84.3 | 82.7 |
| | SID | 84.3 | 84.2 | 89.6 | 89.4 | 86.0 | 85.3 |
| | M3ID | 83.0 | 83.8 | 89.0 | 89.1 | 84.7 | 84.0 |
| | ClearSight | 82.3 | 83.2 | 87.9 | 88.0 | 72.2 | 73.0 |
| | VCD | 83.8 | 83.6 | 88.0 | 88.7 | 85.6 | 83.3 |
| | ICD | 83.3 | 83.2 | 87.8 | 88.4 | 85.8 | 84.7 |
| | OPERA | 83.5 | 82.9 | 88.1 | 89.3 | 86.9 | 84.6 |
| | *CRoPS* | **84.4** | **84.9** | **90.7** | **89.7** | **86.5** | **85.7** |
| Adverserial | Sampling | 76.4 | 78.5 | 82.9 | 83.9 | 82.3 | 81.2 |
| | SID | 77.9 | 79.3 | 84.4 | 84.3 | 83.1 | 83.2 |
| | M3ID | 77.5 | 79.6 | 83.1 | 84.2 | 81.6 | 83.5 |
| | ClearSight | 77.0 | 79.2 | 83.3 | 84.3 | 76.9 | 79.7 |
| | VCD | 77.7 | 79.9 | 83.9 | 84.4 | 82.2 | 81.2 |
| | ICD | 77.1 | 79.2 | 82.3 | 84.3 | 83.0 | 82.1 |
| | OPERA | 77.0 | 79.1 | 84.2 | 84.8 | 82.9 | 82.7 |
| | *CRoPS* | **79.6** | **81.1** | **85.1** | **86.2** | **83.8** | **83.3** |

## H    Qualitative Analysis of Text Tokens Affected by Pruning

We inspect the text tokens ranked by the importance scores in Eq. 5. Figure 6 and 7 visualizes the tokens that receive consistently high importance and are therefore removed in the vision-text-deficit model. Terms such as *left, right, centre, around, within, nearby* and *top* frequently appear among the pruned tokens, showing that the original model depends on explicit positional cues in the prompt. Removing these cues weakens the model's grounding and makes location-related inconsistencies more likely.

A second group contains rare or fine-grained content tokens, including *containing, depth, focus, unusual, expl, transport* and several subword fragments. Although some of them appear only once in our examples, they receive high attention and are pruned frequently. Their removal pushes the model toward more generic language patterns instead of specific details.

The remaining low-importance tokens that survive pruning are mostly weak modifiers or fragments. Overall, the analysis indicates that the vision-text-deficit perturbation removes coherent and informative token groups, rather than random text, which helps explain why it reliably induces hallucination-prone behavior.

## I    Qualitative Examples

Figure 8, 9 and 10 represents the qualitative comparisons between different methods.

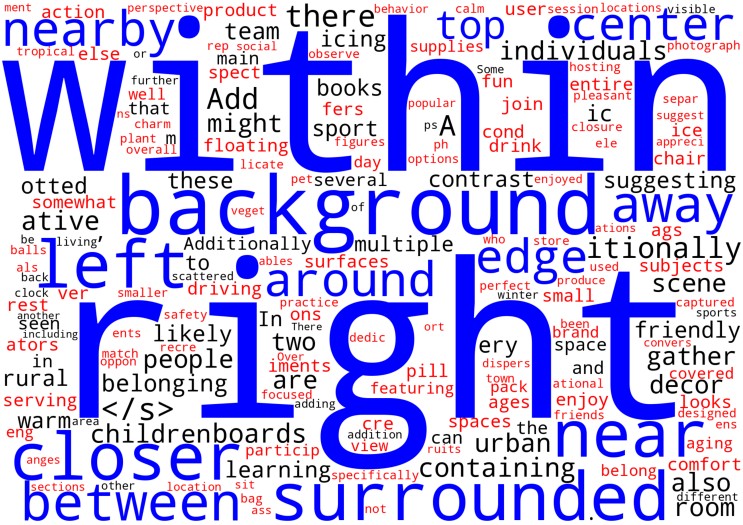

Figure 6: Word cloud of tokens that receive high importance. Positional tokens are blue, rare single-occurrence tokens are red, and all others are black.

The image features a red wall with a water faucet, a vase, and some green plants. The vase is situated next to the faucet, while the plants are placed nearby. There are several bottles scattered around the scene, with one located close to the left edge of the image, another near the center, and the third one further to the right. Additionally ,there is a cup situated near the center of the scene. The combination of the plants, vase, and bottles creates an interesting and somewhat unusual arrangement in the scene.

The image shows a mother goat nursing her baby in a dirt pen. The mother goat is positioned towards the left side of the scene, while the baby goat is standing closer to the center, feeding from its mother. Another adult goat can be seen on the right side of the scene, possibly observing the interaction between the mother and her baby. The scene takes place in an enclosed area, possibly a zoo or farm setting.

Figure 7: **Examples of consistently pruned text tokens**. Highlighted words indicate tokens that receive high importance scores at early layers and are therefore repeatedly removed by the vision-text-deficit perturbation across the entire generation.

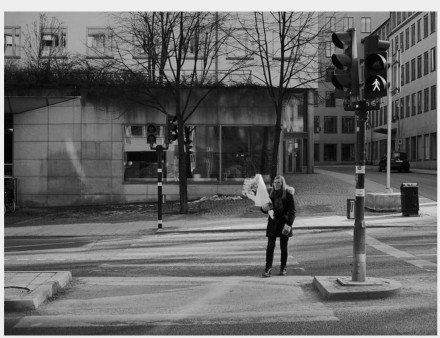

Prompt: Please describe this image in detail

**Vanilla :-** The image depicts a woman holding a white package while crossing the street at an intersection. She is wearing a purple jacket and carrying a handbag. As she is in the process of crossing, it can be inferred that she is looking for the appropriate sidewalk to step onto. The street is equipped with traffic lights and a stop sign, ensuring pedestrian safety. There are multiple traffic lights visible in the scene, with one near the left side, one at the top right corner, and another towards the top left corner of the image. The setting appears to be a residential or suburban area, with a building visible in the background, providing a sense of scale to the scene.

**SID :-** A woman stands on a crosswalk, holding a delivery of papers or white sheet across her body. She is likely waiting for a traffic light to change so she can continue crossing the street. The woman appears to be walking across the intersection as she makes her way. The street is equipped with multiple traffic lights. One traffic light is located in the upper-left corner, another is situated in the lower-left, and two more traffic lights can be seen in the lower-middle section. There is also a handbag on the ground, probably belonging to the woman. The scene appears to be a black and white photo.

**M3ID :-** The image features a woman walking across a street in a city, carrying a large white paper parachute or kite. She is standing in the middle of a crosswalk at an intersection with street lights, waiting to cross. \n\nThere are multiple traffic lights in the scene, with three close to the right side of the intersection and two more in the background. Window coverings can be seen on the buildings on both the left and right sides of the intersection. Her handbag is also visible, placed close to her body while she stands in the crosswalk.

**CRoPS :-** In the image, a woman is standing at an intersection holding a large sheet of paper or a newspaper in her hand. She appears to be crossing the street while navigating through traffic lights. There are multiple traffic lights visible in the scene, some closer to the woman while others are further away. The woman seems to be focused on her task while navigating the urban environment.

Figure 8: Qualitative comparison of generated captions from different methods. Hallucinated words are highlighted in red.

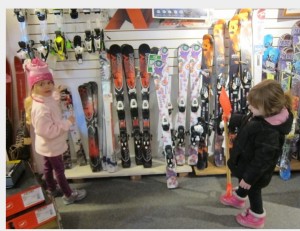

Prompt: Please describe this image in detail

**Vanilla :- The image features two children standing in a snow ski store, surrounded by various skis and snowboards on display. A girl is looking at the skis, while a boy, standing next to her, is curiously examining the snowboards. There are numerous skis in different colors and positions, some closer to the children while others are further away on the right side of the store. A snowboard is also displayed in the background, close to the girl. The children appear to be fully engaged in the winter sports equipment, possibly contemplating which one to choose or simply enjoying the visual variety.**

**SID :- In the store, two little girls are looking at the display of skis and snowboards. The skis are arranged on shelves above the girls, and the snowboards are situated on separate shelves, some placed higher and others lower. The girls are standing close to the display, with one closer to the skis and the other closer to the snowboards. Both girls are wearing black coats, and pink hats are visible on their heads. It appears the girls are browsing the ski shop's offerings, possibly considering which one to purchase for their skiing or snowboarding activities.**

**M3ID :- The image depicts a young girl standing next to a counter filled with various skiing equipment. She is pointing towards a display of skis on the wall, engaging in conversation with another child. Together, they seem to be admiring the store's collection of skis and poles. There are multiple skis throughout the scene, with a particularly large snowboard display among the collection. A smaller skateboard can also be seen. The women at the shop might be mother and daughter, exploring the store and sharing their observation of the skiing gear.**

**CRoPS :- The image features two little girls standing next to each other in a ski shop, admiring the various ski equipment displayed around them. There are multiple pairs of skis visible in the scene, some placed higher up on shelves while others are positioned closer to the ground. The girls seem interested in trying out the skis or learning more about them. One girl appears closer to the left side of the image, while the other girl is positioned more towards the right side of the scene. Both girls seem engaged in the ski shop environment and eager to explore the ski equipment available for purchase or rent.**

Figure 9: Qualitative comparison of generated captions from different methods. Hallucinated words are highlighted in red.

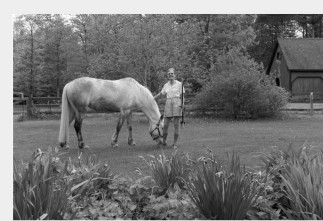

Prompt: Please describe this
image in detail

**Vanilla :-** The image features a woman standing in a grassy field, holding the reins of a horse. **The horse is tied in the field**, **allowing the woman to keep a close eye on it.** A fence can be seen in the background, adding to the rural setting of the scene.

**SID :-** The woman, **who is feeding hay to the white horse in a grassy field**, stands in a black and white picture. She has short hair and appears happy while tending to the horse. The horse stands close to her, displaying a friendly demeanor as she feeds it. In the background, **there are potted plants**, and a fence, possibly in the distance, providing context to the surroundings. Overall, the scene is a calm and peaceful depiction of the woman's connection with her horse.

**M3ID :-** The image portrays a woman standing on a grassy field next to a white horse. The woman is petting the horse, taking the time to engage with it. There is a bridle on the horse's head, indicating that it is a riding horse. **The scene includes a potted plant nearby**, adding a touch of greenery to the lawn area.

**CRoPS :-** The image captures a woman standing in a grassy field next to a white horse. She appears to be petting the horse while holding its bridle in her hand. The scene takes place near some trees, adding a natural element to the environment. The woman appears to be enjoying her time with the horse in this serene setting.

Figure 10: Qualitative comparison of generated captions from different methods. Hallucinated words are highlighted in red.

