# OpenReview forum: "CRoPS: A Training-Free Hallucination Mitigation Framework for Vision-Language Models"
_TMLR — Accepted by TMLR_

### Review · Reviewer_RUwX · 2025-11-09

**Summary Of Contributions:**

The paper tackles the problem of hallucinations in Large Vision-Language Models (LVLMs). While prior training-free approaches attempt to mitigate hallucinations, they often depend on limited assumptions and become less effective toward the end of text generation, where hallucinations are more frequent.

To address these limitations, the authors introduce CRoPS, a training-free hallucination mitigation framework that innovates in two key ways. (1) Text-based hallucinated models – Instead of only removing visual tokens (as done in prior work), the authors selectively remove key text tokens to better simulate hallucination-inducing conditions and capture their effects. (2) Generalized Contrastive Decoding (GCD) – This method integrates multiple hallucinated models to represent diverse sources of hallucination, improving robustness and coverage. The experiments on multiple hallucination benchmarks demonstrate the effectiveness of CRoPS compared with other baseline.

The strengths of this paper include:
- The paper is well-written: the limitation of existing work, the motivation, the proposed methods are clearly stated.
- The experiments are comprehensive: multiple hallucination-related datasets are considered; the authors also consider multiple models and baselines.

However, the paper has several limitations:
- Overall, the paper is incremental given the related work: two improvements have been proposed to better mitigate hallucination, including text-based hallucinated model and a generalized form of contrastive decoding. However, I believe this work is an extension of existing method without notable technical differences. This is acceptable, but, I think more insights are appreciated.
- One major concern is the mechanism of contrastive decoding itself (although it is not the limitation of the paper but rather the background): can you explain why contrastive decoding can help mitigate hallucination as there are many ways of hallucination. Contracting with one type of hallucination can not guarantee the truthfulness of the model. Using multiple hallucinated model seems to be a natural solution. But I do not think this way can mitigate all kinds of hallucinations. Is there any assumption that this type of methods can work?
- How was the hallucinated model trained? The authors can provide more details on the training process.

**Audience:**

Yes

**Audience Explanation:**

Hallucination is an important issue of large multimodal models. The paper tries to address this problem. Thus I believe some individuals are interested in this work.

**Claims And Evidence:**

Yes

**Claims Explanation:**

I believe the technical details presented in this paper are accurate.

**Requested Changes:**

- Please explain the technical novelty of the paper as well as the novel insights. The method is too restricted to a specific kind of hallucination mitigation method, which I think is somewhat narrow (see weakness).
- Please explain the underlying working mechanism of contrastive decoding for hallucination.
- Please specify more training details.

---

> ### Author Response · Authors · 2025-11-28
> **Rebuttal for reviewer RUwX**
>
> We thank the reviewer for the thoughtful feedback. Below we address the concerns point-by-point.
>
> **Q1: Clarification on Technical Novelty and Scope of Our Method**
>
> We appreciate the concern regarding the level of novelty and the scope of insights. In **Section 4 (Motivation)** We provide a detailed analysis explaining why existing contrastive decoding approaches struggle with late-stage hallucinations and with hallucination arising from each modality rather than vision only as studied in earlier literature (Sec. 4.1 to 4.3). This analysis motivates our two main contributions: (i) evidence that using only one hallucinated model is insufficient (Sec. 4.3) and (ii) a vision-text-deficit hallucinated model that removes important textual tokens along with the entire visual tokens, which keeps the contrastive signal effective throughout the entire generation process (Sec. 4.2). Building on these insights, **Section 5 (CRoPS)** introduces a generalized contrastive decoding formulation (Eq. 11) that integrates heterogeneous hallucinated models, one targeting visually induced hallucinations and the other targeting text induced or late stage hallucinations, into a unified decoding strategy. This design enables CRoPS to address multiple sources of hallucination together rather than focusing on a single failure mode.
>
> **Q2: Clarifying the Mechanism and Assumptions of Contrastive Decoding**
>
> Contrastive decoding assumes that if a hallucinated model amplifies a specific bias, then subtracting its logits suppresses that bias in the final output (Section 3.2). CRoPS extends this idea by using multiple complementary hallucinated models that surface the various dominant sources of errors we identify in Section 4, for example visually induced hallucinations in early generation and language-prior-driven hallucinations in later stages.
> We agree that no contrastive decoding method can target every possible hallucination. The practical assumption is that mitigating the major, systematically recurring sources leads to broad improvements. To verify this, CRoPS is evaluated across multiple benchmarks, each designed to capture different hallucination types: CHAIR for object hallucination, AMBER for attribute and relational errors, GPT-4o assisted evaluation for fine-grained object and attribute grounding, and POPE for binary object hallucination. CRoPS achieves consistent gains across all of them and across all LVLM families. This breadth of evaluation demonstrates that combining multiple hallucinated models is effective in mitigating several types of hallucination in practice.
>
> **Q3: Clarifying Training Details of the Hallucinated Model**
>
> We would like to clarify that the hallucinated models in CRoPS are not separately trained models. They are obtained by perturbing the input to the same pretrained LVLM rather than training new networks. Only the input representation is modified, for example
> - For the vision-deficit model, we keep only low-importance vision tokens.
> - For the vision–text-deficit model, we prune vision tokens and high-importance text tokens.
>
> The model parameters remain unchanged; the forward pass is identical except that it receives a perturbed version of the original visual and/or textual tokens. Therefore, no additional training or fine-tuning is involved.

---

### Review · Reviewer_3hM8 · 2025-11-16

**Summary Of Contributions:**

Strengths:
* Methods are well motivated by experimental observations of missing components of existing methods.
* Experimental results and analysis are strong and comprehensive. The paper contains plenty of results to show the effectiveness of the proposed method.

Weakness:
* Despite the theory, the proposed method is mainly adding subtracted logits from input with text-corrupted tokens, which can be seen as a minor modification of existing methods.
* Although evaluation results are promising, due to the lack of benchmarks and access to propriatory models, the evaluation protocol is still somewhat limited. But I understand that this is not the responsibility of authors.

**Audience:**

Yes

**Audience Explanation:**

Overall the method is well motivated with solid results. Despite some minor weakness listed above, I still believe the results are interesting and the method could  be effective in many settings.

**Claims And Evidence:**

Yes

**Claims Explanation:**

As explained in the contribution summarization, the paper in general shows a lot of experimental studies to back their claims and they are satisfying. If I may suggest additional experiments, I would say consider adding another set of experiments on text-only LLMs, since the proposed method is actually applicable there and it would make the paper stronger.

**Requested Changes:**

NA

---

> ### Author Response · Authors · 2025-11-28
> **Rebuttal for reviewer 3hM8**
>
> We thank the reviewer for the thoughtful and constructive comments. Below we address the concern regarding the novelty of our approach.
>
> We agree that CRoPS builds on the contrastive decoding (CD) framework, but it goes beyond simply subtracting logits from a text-corrupted input. As described in both Sec. 4 and Eq. 11, CRoPS introduces a **generalized contrastive decoding formulation** that combines multiple hallucinated models with independent scaling and a time-dependent weighting schedule, which is not supported in prior CD variants.
> In the left subfigure of Fig. 2, we show empirically that **removing visual tokens alone is not sufficient** for modeling hallucinations driven by visual token dilution. As generation progresses, the model’s dependency on visual tokens drops significantly, causing visual-only perturbations to lose contrastive strength. In these later steps, errors propagate into the generated text, and **removing high-importance text tokens becomes more effective** for modeling such hallucinations. Our analysis in Table 7 further demonstrates that a **naive combination of SID and M3ID is not optimal**, highlighting the effectiveness of our proposed design.

---

### Review · Reviewer_nYS9 · 2025-11-22

**Summary Of Contributions:**

The paper presents **CROPS**, a training-free framework for mitigating hallucinations in Large Vision-Language Models (LVLMs).

The key contributions are:
1.  **Motivation & Analysis:** The authors identify a "diminishing visual dependency" phenomenon, where LVLMs rely less on visual tokens as generation progresses. This explains why existing methods (like M3ID or SID) that solely perturb visual inputs lose effectiveness in later generation stages.
2.  **Generalized Contrastive Decoding:** A framework that extends standard contrastive decoding to support multiple negative constraints simultaneously.
3.  **Methodology:** CROPS integrates two distinct "hallucinated" models to target different hallucination sources:
    * A **vision-deficit model** (adapted from SID) that keeps only unimportant visual tokens to capture vision-related errors.
    * A novel **vision-text-deficit model** that removes visual input and retains only unimportant text tokens, specifically targeting language priors that dominate late-stage generation.
4.  **Performance:** Extensive evaluation on three LVLM families (LLaVA-1.5, LLaVA-NeXT, Qwen2-VL) across six benchmarks (including CHAIR, AMBER, POPE) demonstrates consistent SOTA performance, reducing object hallucination by approximately 20% compared to baselines.

**Audience:**

Yes

**Audience Explanation:**

The paper addresses a central challenge in current multimodal AI: reliability and hallucination.
* **Practicality:** As a training-free plug-and-play method, it is immediately applicable to existing open-source models like LLaVA and Qwen, making it highly relevant to practitioners.
* **Theoretical Insight:** The analysis of temporal shifts in visual dependency offers valuable insights into the decoding behavior of autoregressive multimodal models.

**Broader Impact Concerns:**

No structural concerns. The work focuses on mitigating hallucinations, which directly contributes to the safety and reliability of AI systems. No specific Broader Impact Statement is required beyond standard considerations.

**Claims And Evidence:**

Yes

**Claims Explanation:**

The claims are well-substantiated by empirical data:
* **Motivation:** The claim regarding diminishing visual reliance is clearly supported by the Visual Dependency ($VD(t)$) analysis in Figure 2, validating the need for a text-focused intervention in later stages.
* **Effectiveness:** Performance claims are backed by comprehensive comparisons on standard benchmarks. For instance, Table 1 shows CROPS achieving the lowest CHAIR scores across all tested models. Table 4 demonstrates improvements in general perception (MME) and reasoning (MathVista), showing the method does not compromise general capabilities.
* **Design Choices:** Ablation studies in Tables 6 and 7 justify the specific combination of hallucinated models and the token retention policies.

**Requested Changes:**

I recommend the following adjustments to strengthen the submission:

1.  **Discussion of Inference Latency:**
    While Table 5 provides a latency comparison, the text should more explicitly address the computational trade-off. CROPS requires roughly 3x the inference time of standard sampling (652s vs 215s) due to the parallel execution of three models. This significant overhead should be discussed in the main limitations section, along with potential engineering optimizations.

2.  **Hyperparameter Robustness:**
    The framework involves several hyperparameters ($\alpha$, $\gamma$, $\beta$, $\mu$). Table 8 shows sensitivity to these values. The authors should clarify if a single set of hyperparameters was used across all model architectures (LLaVA vs. Qwen) or if model-specific tuning is required. A "default" recommended configuration would enhance reproducibility and usability.

3.  **Qualitative Analysis of Text Pruning:**
    The "vision-text-deficit" model relies on pruning "important" text tokens. Providing qualitative examples of which text tokens are typically classified as "important" (and thus removed) would help intuit why this specific perturbation successfully models language-prior hallucinations.

---

> ### Author Response · Authors · 2025-11-28
> **Rebuttal for reviewer nYS9**
>
> We thank the reviewer for the thorough and insightful evaluation of our work. We appreciate the positive assessment of our motivation, methodology, empirical validation, and relevance to the TMLR audience. Below we address the requested changes point-by-point.
>
> ***Q1: Discussion of Inference Latency?***
>
> Although CRoPS is about 3× slower than vanilla sampling (652s vs 215s), its latency remains comparable to or lower than several existing training-free methods such as VCD (550s), SID (510s), and especially OPERA (1947s). The additional forward pass in CRoPS is lightweight because the vision-text-deficit model processes only a very small subset of tokens rather than a full sequence.
> We have now added a discussion in the **Limitations section** (Highlighted in Blue) related to latency and potential engineering optimizations to mask this latency.
>
> ***Q2: Hyperparameter Robustness?***
>
> We revised the **hyperparameter ablation section** (Highlighted in Blue) to explicitly state the full set of hyperparameters used ($\alpha = 1.0$, $\gamma = 0.02$, $b_0 = 10$, $b_1 = 30$, $\mu = 1\text{e}{-3}$). We also clarify that for all experiments across the LLaVA-1.5, LLaVA-NeXT, and Qwen2-VL backbones and all benchmarks, a single shared set of hyperparameters was used, with no model or data specific tuning.
>
> ***Q3: Qualitative Analysis of Text Pruning***
>
> We appreciate this suggestion. In the revision, we have included qualitative examples in **Appendix H**, illustrating which text tokens are identified as high-importance and therefore removed in the vision–text–deficit model. These examples will help clarify how pruning these tokens exposes the model’s language-prior tendencies and why this perturbation effectively captures late-stage hallucinations.

---

### Decision · Action_Editor_XjBk · 2026-01-01

**Recommendation:** Accept as is

**Audience:**

Yes

**Audience Explanation:**

Hallucination in VLMs are very important in machine learning, especially LLMs and multimodal models's research. The approach is accessible to TMLR readers.

**Claims And Evidence:**

Yes

**Claims Explanation:**

This paper studies the hallucination mitigation in the VLMs, and the authors provide two potential hallucination sources with evidence. The authors also provide the solution to reasonably mitigate these hallucinations.

The experiment are comprehensive across well-known benchmarks. With the review's feedback, the authors extends their experiments and discussions to make to complete.